# Intra-Molecular Homologous Recombination of Scarless Plasmid

**DOI:** 10.3390/ijms21051697

**Published:** 2020-03-02

**Authors:** Yaping Liang, Yu Zhang, Liangwei Liu

**Affiliations:** 1The Life Science College, Henan Agricultural University, Zhengzhou 450002, China; 15838021692@163.com (Y.L.); zhang_yu_0919@163.com (Y.Z.); 2The Key Laboratory of Enzyme Engineering of Agricultural Microbiology, Ministry of Agriculture, Henan Agricultural University, Zhengzhou 450002, China

**Keywords:** intra-molecular, homologous recombination, scarless, plasmid

## Abstract

Although many methods have been reported, plasmid construction compromises transformant efficiency (number of transformants per ng of DNAs) with plasmid accuracy (rate of scarless plasmids). An efficient method is two-step PCR serving DNA amplification. An accurate method is ExnaseII cloning serving homology recombination (HR). We combine DNA amplification and HR to develop an intra-molecular HR by amplifying plasmid DNAs to contain homology 5′- and 3′-terminus and recombining the plasmid DNAs in vitro. An example was to construct plasmid pET20b-*AdD*. The generality was checked by constructing plasmid pET21a-*AdD* and pET22b-*AdD* in parallel. The DNAs having 30-bp homology arms were optimal for intra-molecular HR, and transformation of which created 14.2 transformants/ng and 90% scarless plasmids, more than the two-step PCR and the ExnaseII cloning. Transformant efficiency correlated with the component of nicked circular plasmid DNAs of HR products, indicating nick modification in vivo leads to scar plasmids.

## 1. Introduction

As a foundational technology in molecular biology, plasmid construction is essential for investigating the functions of DNA and protein [1,2]. Although many cloning methods have been reported, plasmid construction compromises transformant efficiency (i.e., number of transformants per ng of plasmid DNAs used in transformation) with plasmid accuracy (i.e., rate of scarless plasmids). The type II cloning method is limited by low activity of restriction enzymes, insertion of restriction sites, and specific positions for insertion [3]. These drawbacks drive DNA amplification methods to serve amplified inserts as megaprimers [2,4,5,6,7,8]. One method is two-step PCR extending megaprimer along with vector to amplify plasmid DNAs, ligating the DNAs in blunt-end by T4 DNA ligase, and transforming the ligaed DNAs into host cells [2] (Figure 1, middle). Although efficient for creating over 10 transformants/ng, the method is compromised by constructing occasionally scar plasmids, i.e., insert-vector junctions contain nucleotide deletion, insertion, or mutation. Scar junction is a general problem in plasmid construction [4,5,7,9,10,11,12,13,14], and a frame-shift scar makes functional analysis impossible.

Different from DNA amplification methods, ET cloning is more accurate by serving homologous recombination (HR) in vivo [15,16]. However, the in vivo HR is inconvenient by co-transforming inserts and vectors having homology arms, recovering recombined plasmids from clone cells, and re-transforming them into expression cells. A relatively convenient method is the ExnaseII cloning (Vazyme Biotech, Nanjing, China). The in vitro HR recombines inserts and vectors having homology arms by recombinase ExnaseII, and transforms HR products directly into expression cells (Figure 1, right). However, both methods serve intermolecular HR, which compromises transformant efficiency.

Accordingly, we combine the two-step PCR and HR to develop an intra-molecular HR (Intra-HR) (Figure 1, left). An example is to construct plasmid pET20b-*AdD* from the template pTrc-*DBD*-*AdD* containing DNA-binding domain and adenylation domain (residue 130~487) of T4 DNA ligase [17], which uses widely in DNA ligation [17,18,19]. The generality is checked by constructing plasmid pET21a-*AdD* and pET22b-*AdD* in parallel. The intra-HR compares with the two-step PCR and the ExnaseII cloning for transformant efficiency and plasmid accuracy. The optimal length analyzes for homology arm. Moreover, component of HR products is determined for which to create transformants, and the reason analyzes for scar plasmids.

## 2. Results

### 2.1. Intra-HR into Plasmid

#### 2.1.1. Protocol Overview

The intra-HR needs three chimeric primers: AFT, T4R, and VRPnT (Figure 1, left), with the first two amplifying the insert, the 1st and the 3rd providing the intramolecular homology arms. A 1st-PCR serves the first two primers to amplify the insert *AdD* from the template pTrc-*DBD-AdD*. Coupling with the primer VRPnT, a 2nd PCR serves the amplified insert as a megaprimer to extend along with the vector to amplify plasmid DNAs of pET20b-*AdD*. The amplified plasmid DNAs have homology 5′- and 3′-termini provided by the primer 1st and 3rd complementary 5′-termini. After intramolecular HR by the commercial recombinase ExnaseII (Vazyme Biotech), the plasmid DNAs transform into *E. coli* cells [3].

#### 2.1.2. Primer Design Rule

The primer AFT has 40 nucleotide 5′-terminus complementing with the primer VRPnT 5′-terminus, and 30 nucleotide 3′-terminus identifying to the insert 5′-terminus. Serving the amplified insert as forward megaprimer and determining the end of insertion, the primer T4R has 24 nucleotide 5′-terminus complementing with the vector at the insertion downstream, and 24 nucleotide 3′-terminus complementing with the insert 3′-terminus. Determining the start of insertion, the primer VRPnT has 40 nucleotide 5′-terminus complementing with the primer AFT 5′-terminus, and 35 nucleotide 3′-terminus complementing with the vector at the insertion upstream.

#### 2.1.3. Plasmid Amplification, Recombination, and Transformation

The amplified megaprimer AdD was at the expected band 1.1 kb (Figure 2, Lane (L) 3-4), consistent with the 358 residue-sized domain. The amplified plasmid DNAs were at the expected band 4.8 kb (Figure 2, L6), consistent with the plus of the 1.1 kb insert and the 3.7 kb vector. After intra-molecular HR in vitro, the plasmid DNAs transformed into *E. coli* BL21(DE_3_) competent cells in the chemical method [3].

#### 2.1.4. Transformant Efficiency and Plasmid Accuracy

The transformant efficiency was six transformants/ng (Table 1). Screening 10 transformants found seven expectation-sized plasmids and three empty-sized vectors (Figure 3. 1′L6, 9-10). Sequencing found five of the seven expectation-sized plasmids having scarless junctions (Figure 3. 1′L1-2, 4-5, 7) and two having scar junctions (1′L3, 8), i.e., 50% were scarless plasmids. The scar junctions deleted respectively nucleotide +16C and +22C (with the number and alphabet shown as the position of deleted nucleotide relative to the scarless junction).

#### 2.1.5. The Intra-HR Generality

In amplifying the plasmid DNAs of pET20b-*AdD*, the megaprimer *AdD* and the primer VRPnT annealed respectively to the vector His-tag and ribosome-binding site (RBS), two general regions of the expression vector pET21a and pET22b. Thereby, the method generality was checked by changing vector to pET21a or pET22b. Amplified DNAs were at the expected 6.5 kb band, equaling to the plus of the 1.1 kb insert and the 5.4 kb vector pET21a or pET22b (Figure 2, L8, 10).

After intra-molecular HR and transformation, the plasmid DNAs created 9.8 transformants/ng for pET21a-*AdD* and 37.3 for pET22b-*AdD* (Table 1). Screening 10 transformants found eight expectation-sized plasmids for pET21a-*AdD* and nine for pET22b-*AdD*. The rest were two dimer-sized plasmids for pET21a-*AdD* and 1 for pET22b-*AdD*. Sequencing found all eight and nine of the expectation-sized plasmids having scarless junctions, i.e., 80% were scarless plasmids for pET21a-*AdD* and 90% for pET22b-*AdD* (Table 1). The three dimer-sized plasmids originated probably from inter-molecular HR of plasmid DNAs.

### 2.2. Comparing with the Two-Step PCR

In parallel with the above intra-HR, the two-step PCR amplified ligation-for plasmid DNAs of pET20b-*AdD* (Figure 2, L5). After ligation and transformation, the plasmid DNAs created 2.1 transformants/ng. Screening 10 transformants found seven expectation-sized plasmids (Figure 3. 2′L2-3, 6-10). However, sequencing found all seven of the plasmids having scar junctions, with the first four deleting nucleotide -1G and the latter three deleting -4T-3A-2T-1G, -2T-1G, or -3A-2T-1G respectively (with the number showing the position of the deleted nucleotide relative to the scarless junction). In contrast, the intra-HR created six transformants/ng, 50% of which had scarless plasmids (Table 1).

### 2.3. Comparing with the ExnaseII Cloning

In another parallel with the intra-HR, the ExnaseII cloning served for inter-molecular HR. As to the DNAs having 20-bp homology arms, the ExnaseII cloning created a similar plasmid accuracy (80% vs. 80%) but a 2-fold fewer transformant efficiency than the intra-HR (5.7 vs. 9.7 transformants/ng) (Figure 4 and Table 1). As to the DNAs having 40-bp homology arms, the ExnaseII cloning created a similar transformant efficiency (13.5 vs. 13.7 transformants/ng) but a 3-fold fewer plasmid accuracy than the intra-HR (30% vs. 90%). In a word, the intra-HR created 2~3-fold more scarless plasmids. In contrast, the ExnaseII cloning produced six scar junctions at the joint 5′-terminus and one at the 3′-terminus. Theoretically, the ExnaseII cloning serves two enzyme molecules for inter-molecular HR of two joints, whereas the intra-HR serves one molecule for intra-molecular HR of one joint (Figure 1).

### 2.4. Optimal Length of Homology Arm

To analyze optimal length of homology arm, we amplified plasmid DNAs of pET20b-*AdD* having 10-, 30-, or 50-bp homology arms. After intra-molecular HR and transformation, the DNAs having 30-bp homology arms created 14.2 transformants/ng (Figure 5 and Table 1), and 90% of which had scarless plasmids. The DNAs having 10- or 50-bp homology arms created respectively 12.7 or 11.4 transformants/ng, and 60% or 80% of which had scarless plasmids. Therefore, the DNAs having 30-bp homology arms were optimal for intra-molecular HR.

### 2.5. Origin of Transformant

To determine which component of HR products create transformants, we separated HR products and ligation products by gel-electrophoresis. Components were monomer, nicked circular plasmid (nc-plasmid), and multimer DNAs (Figure 6A). However, intact plasmid DNAs were not there lower than the monomer DNAs. Each component was quantified by Gel-Pro analyzer software, and correlated with transformant efficiency. A correlation had only with the component of nc-plasmid DNAs, and not that of whole, or monomer DNAs (Figure 6B). The correlation equation was Y = −6.03 + 1.08 × X (with Y and X representing respectively transformant efficiency and quantity of nc-plasmid DNAs). Theoretically, transformation needs at least 5.6 ng of nc-plasmid DNAs. Similar to DNA ligation [20,21], nc-plasmid DNA formation depends on joints, HR efficiency, and DNA purity. Transformants originate from the component of nc-plasmid DNAs. Scar junctions are attributed to nick modification of the nc-plasmid DNAs by host cells.

## 3. Discussion

### 3.1. Advantages

The intra-HR method has three advantages: (1) plasmid accuracy. The intra-HR method constructed 50% scarless plasmids for pET20b-*AdD*, 80% for pET21a-*AdD*, and 90% for pET22b-*AdD*. In contrast, the two-step PCR constructed zero scarless plasmids for pET20b-*AdD*. The ExnaseII cloning constructed 2~3-fold fewer scarless plasmids for pET20b-*AdD*. Scarless junction is the first demand in plasmid construction, especially for protein expression. Frame-shift junction makes impossible for analyzing DNA function, using RBS, or serving His-tag purification. The demand for scarless junction drives development for seamless cloning [22], and (2) transformant efficiency. The Intra-HR can clone simultaneously an insert into similar vectors as pET20b, pET21a, or pET22b, because the vectors contain general regions as RBS and 6His-tag. Transformant efficiency was 6~37 transformants/ng into home-made competent cells *E. coli* BL21(DE_3_) in chemical method, significantly higher than the ET cloning, which needs electro-transformation [23]. HR relies on thermal motion for two joint meetings. 5′- and 3′-terminus meet more easily for one than two DNA molecules, which need to align sequentially and recombine simultaneously. Thereby, it is more efficient and accurate for intra- than inter-molecular HR, and (3) free of restriction enzymes and DNA ligase. Similar to other DNA amplification methods [4,5,9,10,11,12,13], the intra-HR can clone an insert into any position of a vector without the limitation of endonuclease and free of restriction sites. Determining the start and the end of insertion, the primer T4R and the primer VRPnT complement respectively with the vector at downstream or upstream of the insertion site.

### 3.2. Limitations

The intra-HR has two limitations: (1) the commercial enzyme ExnaseII is unknown, and thereby unclear is the mechanism. The ExnaseII has a similar molecular weight with the proteins RecE588/T, indicating the former might be similar to the latter [24]. A similar cell extract serves for in vitro HR in the SLiCE method [25]. The ExnaseII cloning created scar junctions deleting nucleotides mainly at the joint 5′-terminus, consistent with the RecE588 having 5′-exonuclease activity [24]. Differently, the two-step PCR created scar junctions deleting nucleotides mainly at the joint 3′-terminus, consistent with DNA ligase disassembling double- into single-strand and changing the 3′-joint into A-like conformation [24,26], and (2) the intra-HR can clone a single insert, fewer than several inserts that of the ExnaseII cloning and the ET cloning. The intra-HR can use megaprimer long as to 2 kb, although amplification efficiency decreases with increasing length, similar to other DNA amplification methods [4,6].

### 3.3. Scar-Producing Mechanism

Although we can screen scarless plasmids from different transformants, scar junction is headache in plasmid construction. Because scar plasmids usually occur [5,7,8,12,13,14,22], plasmid construction has gone a long way from free of restriction sites, seamless cloning, to scarless cloning. Plasmid construction involves two steps as construction and transformation. However, investigation concentrates mainly on the 1st step [5,12,22]. We find that almost all methods construct nc-plasmid DNAs, such as type II cloning, Gibson [27], SLiCE [25], the DNA amplification methods [4,6,9,10,11], the ExnaseII cloning, and the intra-HR. The nicks, overhangs, or gaps of nc-plasmid DNAs need search, anneal, and repair by host cell resolvases, nucleases, and polymerases [6,13,14]. One is the *S. cerevisiae* exonuclease EXO1 to resect DNA ends into 3′-overhangs, one essential intermediate for double-stranded break repair [28]. The formation of nc-plasmid DNAs is an abortive ligation of double-stranded break [29]. Nick modification of nc-plasmid DNAs in vivo is a kind of double-strand break repair [30,31]. The process might delete, insert, or mutate nucleotides at the joint, and thereby, produce scar inter-vector junctions.

Seen from the above data, the intra-HR amplifies plasmid DNAs having 10 to 50 bp homology 5′ and 3′-termini (Figure 1, left). ExnaseII recombines the plasmid DNAs into stable nc-plasmid DNAs for having 10 to 50 bp complementary strand, and nicks are more likely ligated by host cells after transformation. The result is higher transformant efficiency and plasmid accuracy. On the other hand, the two-step PCR amplifies plasmid DNAs having blunt-ended 5′ and 3′-termini (Figure 1, middle). T4 DNA ligase ligates the plasmid DNAs into unstable nc-plasmid DNAs for having 0 bp complementary strand, and nicks of the nc-DNAs are more likely modified by host cells after transformation. The result is lower transformant efficiency and plasmid accuracy.

## 4. Materials and Methods

### 4.1. Materials

The regents were Q5 DNA polymerase, T4 DNA ligase, DpnI (NEB, Beijing, China), and ExnaseII (Vazyme Biotech, Nanjing, China). The template plasmid pTrc-*DBD-AdD* was kindly provided by Alessandra Montecucco [17]. Primers synthesized for AFT, T4R, VRPnT, AF, VRPn, AF-L, AR-L, ART, VF, VFT, ARP1, ARP3, and ARP5 (Sangon Biotech, Shanghai, China) (Table 2). The primers AFT and VRPnT had 40 bp complementary 5′-temini (underlined in straight line). The primer T4R 5′-terminus complements with the vector pET20b for 24 nt (underlined in wavy line).

### 4.2. Intra-HR of Plasmid pET20b-AdD

#### 4.2.1. Megaprimer Amplification

The AFT and T4R primers served to amplify megaprimer *AdD* (1.1 kb) from the template pTrc-*DBD-AdD*. The PCR reagents were 200 nM AFT and T4R primers, 3.6 ng pTrc-*DBD-AdD* templates, 1 U Q5 DNA polymerase, and 1 × Q5 DNA polymerase buffer. The PCR procedures were pre-denaturation at 98 °C for 3 min, 30 cycles of denaturation at 98 °C for 30 s, double-annealing at 60 °C for 10 s and 58 °C for 10 s [32], extension at 72 °C for 50 s, and extension at 72 °C for 10 min. The DNAs purified from a 0.8% gel by DNA clean kit (Axygen, Corning, NY, USA), and assayed concentration by Nanodrop 1000 (Thermo Scientific, Shanghai, China).

#### 4.2.2. Plasmid Amplification

Coupled with the primer VRPnT extending along with the respective vector, a 2nd PCR served the megaprimer *AdD* to amplify plasmid DNAs of pET20b-*AdD*, pET21a-*AdD*, and pET22b-*AdD*. The PCR reagents were similar except for 85 ng *AdD* megaprimers, 200 nM primer VRPnT, and 37.8 ng pET20b, 35.2 ng pET21a, or 20.2 ng pET22b templates, respectively. The PCR procedures were similar except for double-annealing at 65 °C for 10 s and 58 °C for 10 s, and extension at 72 °C for 2 min 30 s. Megaprimer annealing temperature (T_m_) calculated by an empirical equation: T_m_ = 70.5 − kb × 7.46, with the kb as the length of non-complementary nucleotides with the template.

#### 4.2.3. Intra-Molecular HR and Transformation

A 25 µL HR volume contained 100 ng linear DNAs of pET20b-*AdD*, pET21a-*AdD*, or pET22b-*AdD*, 1 × CE II buffer, and 2 µL ExnaseII (U not shown in the user manual). HR carried out at 37 °C for 30 min and stopped at 0 °C for 5 min. HR products containing 30 ng linear plasmid DNAs transformed into 200 µL home-made competent cells *E. coli* BL21(DE_3_) at 42 °C [3]. Transformant efficiency calculated as an average of triplicates for number of transformants per ng of linear plasmid DNAs used for transformation. Electrophoresis separated the HR products into different components, and each component quantified by Gel-Pro Analyzer 4.0 (Media Cybernetics, Bethesda, MD, USA). Correlation made between quantity of nc-plasmid DNAs and transformant efficiency (OriginLab 8, Northampton, MA, USA). We screened 10 transformants to find expectation-sized plasmids, insert-vector junctions of which were checked scarless or not by DNA sequencing (Sangon Biotech, Shanghai, China).

### 4.3. Comparing with the Two-Step PCR

The AF and T4R primers served to amplify ligation-for megaprimer *AdD*’ from the template pTrc-*DBD-AdD*. Coupled with the primer VRPn extending along with the vector pET20b, a 2nd PCR served the megaprimer *AdD*’ to amplify plasmid DNAs of pET20b-*AdD*, ligation-for ends of which were from the phosphorylated AF and VRPn primers. Ligation carried at 16 °C for 16 h in a 30 µL volume containing 100 ng linear pET20b-*AdD* DNAs, 400 U T4 DNA ligase, and 1 × T4 DNA ligase buffer. After digesting templates by 20 U DpnI, the ligation products transformed in a similar method.

### 4.4. Comparing with the ExnaseII Cloning

#### 4.4.1. Inter-Molecular HR

The primer pair AFT/ART or AF-L/AR-L served to amplify insert *AdD* DNAs having 40- or 20-bp inter-molecular homology arms with vector pET20b DNAs amplified by the primer pair VF/VRPn or VFT/VRPnT. The *AdD*-amplification reagents were similar except for 32.4 ng pET20b-*AdD* and 500 nM primer pair AF-L/AR-L or AFT/ART. The *AdD*-amplification procedures were similar except for double-annealing at 72 °C for 15 s and 57 °C for 15 s, and extension at 72 °C for 30 s. The vector-amplification reagents were similar except for 49 ng pET20b templates and 500 nM primer pair VF/VRPn or VFT/VRPnT. The vector-amplification procedures were similar except for double-annealing at 72 °C for 15 s and 61 °C for 15 s, and extension at 72 °C for 2 min 30 s. The HR reagents were similar except for 58 ng insert DNAs and 173 ng linear vector DNAs.

#### 4.4.2. Intra-Molecular HR

The AFT and ART primers served to amplify megaprimer *AdD* from the template pET20b-*AdD*. The *AdD*-amplification reagents were similar except for 32.4 ng pET20b-*AdD* templates and 500 nM AFT and ART primers. The *AdD*-amplification procedures were similar except for double-annealing at 72 °C for 10 s and 57 °C for 15 s, and extension at 72 °C for 10 s. Coupled with the primer VRPn or VRPnT, a 2nd PCR served the megaprimer to amplify plasmid DNAs having 20- or 40-bp intra-molecular homology arms. The plasmid DNA-amplification reagents were similar except for 210 ng megaprimers *AdD*, 32.4 ng templates pET20b, and 500 nM 20- or 40-bp homology-for primer VRPn or VRPnT. The plasmid DNA-amplification procedures were similar except for double-annealing at 67 °C for 10 s and 61 °C for 15 s, and extension at 72 °C for 2 min 20 s. The HR regents were similar except for 200 ng plasmid DNAs having 20- or 40-bp homology arms. Because the 4.8 kb intra-HR-for DNAs were larger than the 3.7 kb inter-HR-for DNAs, equivalent vectors calculated for transformant efficiency.

### 4.5. Optimal Length of Homology Arm

To analyze optimal length of homology arm for intra-molecular HR, the primer VFT and ARP1, ARP3, or ARP5 served to amplify plasmid pET20b-*AdD* DNAs having 10-, 30-, or 50-bp homology arms. The PCR reagents were similar except for 64.8 ng pET20b-*AdD* templates, 500 nM primer VFT, and 500 nM primer ARP1, 5 nM primer ARP3, or 5 nM primer ARP5. The PCR procedures were similar except for annealing at 72 °C 20 s, and extension at 72 °C for 2 min 30 s. The HR reagents were similar except for 200 ng plasmid DNAs having 10-, 30-, or 50-bp homology arms. After digesting templates by 20 U DpnI, the HR products transformed in a similar method.

## 5. Conclusions

To summarize, plasmid construction compromises transformant efficiency with plasmid accuracy. Combining the two characters, we develop an intra-HR to amplify plasmid DNAs containing homology 5′- and 3′-terminus. After intramolecular HR in vitro by ExnaseII at 37 °C for 30 min, the plasmid DNAs transform into expression cells *E. coli* BL21(DE_3_). The intra-HR constructed plasmid pET20b-*AdD*, pET21a-*AdD*, and pET22b-*AdD* in parallel. The DNAs having 30 bp homology arms were optimal for intra-molecular HR, transformation of which created 14.2 transformants/ng and 90% scarless plasmids. In contrast, the two-step PCR constructed zero scarless plasmids, and the ExnaseII cloning constructed 2~3-fold fewer scarless plasmids. The component of nc-plasmid DNAs correlated with transformant efficiency. Transformation of nc-plasmid DNAs leads to nick modification in vivo, and the result is scar plasmids.

## Figures and Tables

**Figure 1 ijms-21-01697-f001:**
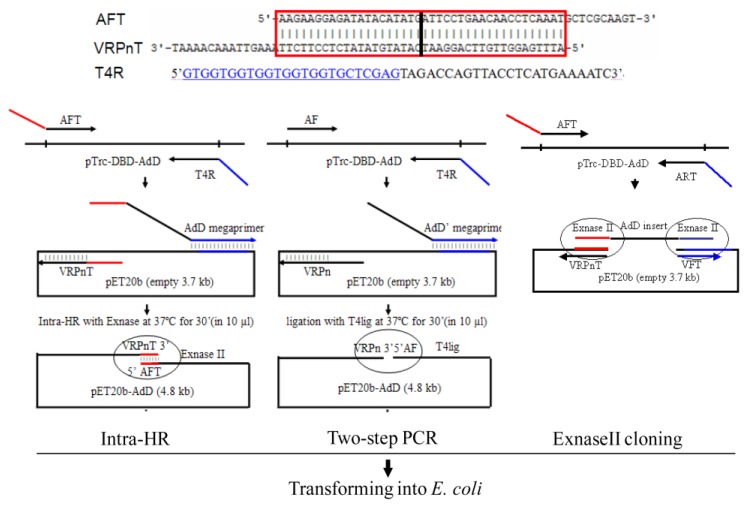
Protocol of the intra-HR (left), the two-step PCR (middle), and the ExnaseII cloning (right). The AFT and VRPnT primers in the intra-HR (left) have complementary 5′-termini (red line), differing from the un-complementary primers of AF and VRPn in the two-step PCR (middle). The primer T4R in both the intra-HR (left) and the two-step PCR (middle) complements to the vector at the insertion downstream (blue line), providing the amplified megaprimer to anneal to the vector in a 2nd PCR. Coupled with the primer VRPnT (left) or the primer VRPn (middle), the megaprimer *AdD* (left) or the megaprimer *AdD*’ (middle) amplify linear plasmid DNAs to have homology (left) or blunt-ended 5′- and 3′-terminus (middle). The intra-molecular homology region contains the expected junction of the vector 3′-end and the insert 5′-end (vertical straight black line in the red box). In the ExnaseII cloning (left), the AFT and ART primers amplify the insert *AdD* having homology arms (red and blue lines) with the vector amplified by the VFT and VRPnT primers. The intra-HR serves one ExnaseII molecule for intra-molecular HR of one joint (left), differing from two ExnaseII molecules for inter-molecular HR of two joints that of the ExnaseII cloning (right). The plasmid DNAs recombine by recombinase ExnaseII (left and right) or ligate by T4 DNA ligase (middle), and transform into *E. coli* cells.

**Figure 2 ijms-21-01697-f002:**
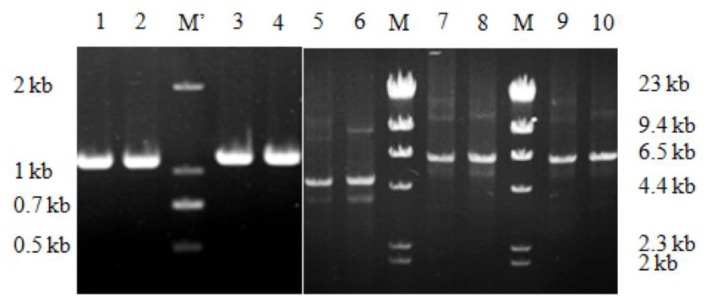
Gel electrophoresis of the amplified megaprimers and the plasmid DNAs. DL2000 DNA ladder (M’) and HindIII-digested λ-DNA (M) used as marker. The intra-HR-for megaprimer *AdD* (L3-4) and the two-step PCR-for megaprimer *AdD*’ (L1-2) were both at the expected 1.1 kb bands. The intra-HR-for plasmid DNAs were at the expected 4.8 kb for pET20b-*AdD* (L6), 6.5 kb for pET21a-*AdD* (L7-8), and 6.5 kb for pET22b-*AdD* (L9-10). The two-step PCR-for plasmid DNAs were at the expected 4.8 kb for pET20b-*AdD* (L5).

**Figure 3 ijms-21-01697-f003:**
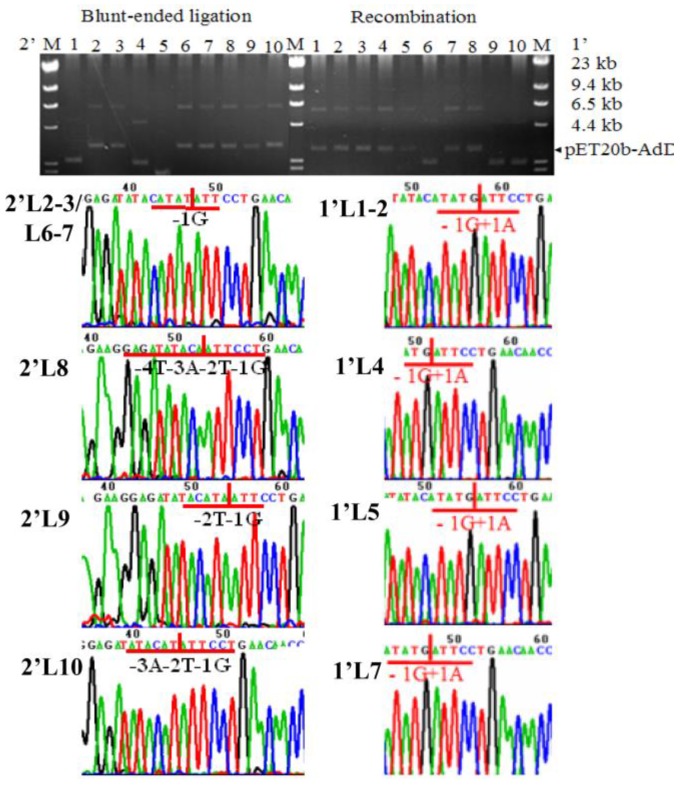
Plasmid analysis. Each selected 10 transformants for the intra-HR (1′L1-10, right) and the two-step PCR (2′L1-10, left). Seven plasmids from the intra-HR had expected sizes (right). Five of the seven expectation-sized plasmids had scarless junctions (1′L1-2, 4-5, 7) (with vertical straight line, number, and alphabet in red showing the position and nucleotide of the scarless junction -1G+1A). Seven plasmids from the two-step PCR had expected sizes (left). All seven of the expectation-sized plasmids had scar junctions, with nucleotide -1G deleted in the 2′L2, 3, 6, and 7, -4T-3A-2T-1G in the 2′L8, -2T-1G in the 2′L9, and -3A-2T-1G in the 2′L10 (with vertical straight red line, black number and letter shown as the position of the deleted nucleotide).

**Figure 4 ijms-21-01697-f004:**
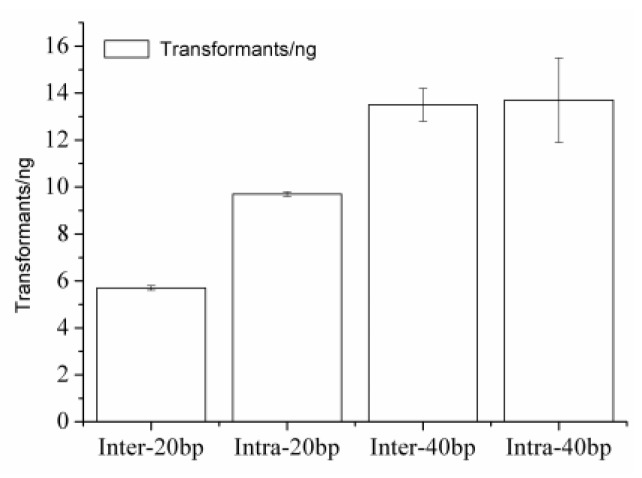
Comparing the ExnaseII cloning with the intra-HR. Intra-20 bp, -40 bp: transformant efficiency of intra-HR-for DNAs having 20 bp or 40 bp homology arms, inter-20 bp, -40 bp: transformant efficiency of ExnaseII cloning-for DNAs having 20 bp or 40 bp homology arms.

**Figure 5 ijms-21-01697-f005:**
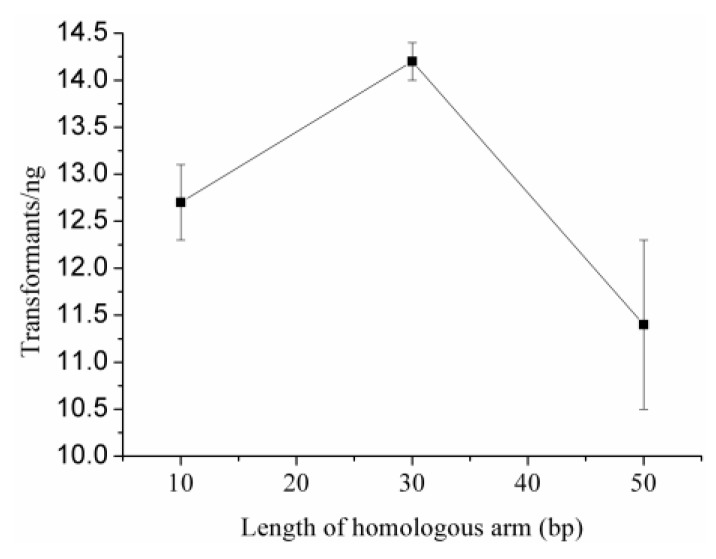
Optimal length analysis of homology arms. The DNAs having 30-bp homology arms created higher transformant efficiency and plasmid accuracy than those having 10- and 50-bp homology arms.

**Figure 6 ijms-21-01697-f006:**
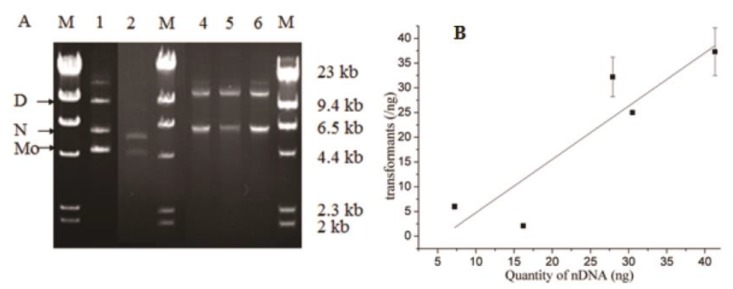
Analysis of nicked circular plasmid (nc-plasmid) DNAs (**A**) and correlation with transformants (B). (A) Gel-electrophoresis separated the HR products of pET20b-*AdD* (L2), pET21a-*AdD* (L4), and pET22b-*AdD* DNAs (L6) into component as monomer (Mo), nicked circular (N), and dimer DNAs (D). Similar components separated for the ligation products of pET20b-*AdD* (L1) or pET22b-*AdD* (L5). Different components were quantified by Gel-Pro analyzer software. (**B**) Correlation analyzed for quantity of nc-plasmid DNAs with transformant efficiency by Origin software. An equation was: Y = −6.03 + 1.08 × X (error bars represent standard deviation, Adj. R-Square 0.74).

**Table 1 ijms-21-01697-t001:** Summary of the intra-HR, the two-step PCR, and the ExnaseII cloning.

Method	Enzyme/Reaction	Plasmids	Efficiency (Transformants/ng)	Nicked DNA (ng)	Accuracy (%)/Scarless Plasmids
Intra-HR in parallel with the two-step PCR	ExanseII 37°C, 30 min/intra-molecular HR	pET20b-*AdD* (1′ in Figure 3)	6 ± 0.5	7.2	50/1′L1-2,4-5,7
		pET21a-*AdD*	9.8 ± 1.5	14.1	80/L1-4/7-10
		pET22b-*AdD*	37.4 ± 6.8	41.3	90/L1-4/6-10
Two-step PCR	T4 DNA ligase 16°C, 16 h/intra-molecular ligation	pET20b-*AdD* (2′ in Figure 3)	2.1 ± 0.5	16.2	0/
		pET21a-*AdD*	25 ± 0.1	30.5	50/L2,4,6-7,9
		pET22b-*AdD*	32.2 ± 5.7	27.9	70/L1-4,8-10
ExnaseII cloning	ExanseII 37°C, 30 min/inter-molecular HR	pET20b-*AdD* with inter-20bp	5.7 ± 1.1	nd	80
		pET20b-*AdD* with inter-40bp	13.5 ± 2.7	nd	30
Intra-HR in parallel with the ExnaseII cloning	ExanseII 37°C, 30 min/intra-molecular HR	pET20b-*AdD* with intra-20bp	9.7 ± 1.2	nd	80
		pET20b-*AdD* with intra-40bp	13.7 ± 2.3	nd	90
Intra-HR for optimal arm length	ExanseII 37°C, 30 min/intra-molecular HR	pET20b-*AdD* with intra-10bp	12.7 ± 0.4	nd	90
		pET20b-*AdD* with intra-30bp	14.2 ± 0.2	nd	60
		pET20b-*AdD* with intra-50bp	11.4 ± 0.9	nd	80

Note: HR: homologous recombination, Efficiency: number of transformants per ng of linear plasmid DNAs used for transformation, Accuracy: rate of scarless plasmids in 10 transformants, nd: not detected, transformant efficiency was an average of triplicate transformations with each having 30 ng of linear plasmid DNAs. 10 transformants selected for each of the 3 methods. Plasmids having expected sizes were sequenced to check junctions scarless or not. Quantity of nicked plasmid DNAs were analyzed by Gel-Pro Analyzer 4.0.

**Table 2 ijms-21-01697-t002:** Primer sequences and melting temperatures.

Primer	Sequence	T_m_ (°C)
AFT	AAGAAGGAGATATACATATG‖ ATTCCTGAACAACCTCAAATGCTCGCAAGT	72
T4R	GTGGTGGTGGTGGTGGTGCTCGAGTAGACCAGTTACCTCATGAAAATC	61
VRPnT	ATTTGAGGTTGTTCAGGAAT‖ CATATGTATATCTCCTTCTTAAAGTTAAACAAAAT	61
AF	p-ATTCCTGAACAACCTCAAATG	72
VRPn	p-CATATGTATATCTCCTTCTTAAAGTTAAACAAAAT	62
AF-L	ATTCCTGAACAACCTCAAATGCTCGCAAGT	72
AR-L	TTCTTTAAATTTATAAAGATTTTTTGAACG	57
ART	TGGTGGTGGTGGTGCTCGAGTTCTTTAAATTTATAAAGATTTTTTGAACG	57
VF	CTCGAGCACCACCACCACC	65.2
VFT	ATCTTTATAAATTTAAAGAACTCGAGCACCACCACCACCA	73
ARP1	GTTCAGGAATCATATGTATATCTCCTTCTTAAAGTTAAACAAAATTATTTCTAGA	68
ARP3	ACTTGCGAGCATTTGAGGTTGTTCAGGAATCATATGTATATCTCCTTCTTAAAGT	75
ARP5	ATGCCTTTTTCATCATAAGAACTTGCGAGCATTTGAGGTTGTTCAGGAATCATAT	76

Note: “‖” indicating intentional junction of the vector 3′-OH and the insert 5′-PO_4_.

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
