# Peer review of "Intra-Molecular Homologous Recombination of Scarless Plasmid"

_ijms, 2020, doi:10.3390/ijms21051697_

Round 1

Reviewer 1 Report

The article describes a novel method for DNA amplification that is less erroneous in terms of nucleotide mutation, insertion, and deletion - collectively referred to as ‘scaring’ - but at the expense of recombination efficiency as a function of mixture concentration and sequence length. While the idea is interesting and the implementation legitimate, the paper severely lacks in (1) technical prose and clarity, (2) structures, including but not limited to schematics, flowcharts, and tables, for the many experimental procedures, many of details are similar or repeated, (3) high-resolution figures with appropriate spacing and contrast, and (4) discussions of the broader significance of the method within the commercial, academic, and healthcare communities. 

The article shows promise that should be realized by giving it more thought to the four critical elements mentioned above, and thus, I do not as yet recommend this paper for publication. 

Below are a number of suggestions on how the article may be improved upon

(1) Technical prose

Why is a particular DNA sequence, pET20b-AdD chosen for the experiment?

What is specifically “convenient” or “inconvenient” about the different methods?

Define efficiency and accuracy when they are first mentioned.

Introduction should include current state-of-the-art DNA amplification method and with references to better define the relevance of the article

Conclusion should continue where introduction ends. 

Description of procedures, results and discussions should be clearly connected to each other even as they are segmented. These are not so in the article. Additionally, each section should flow from one to the next

Pros and cons of each method are better summarized in the form of a table. 

(2) Essential schematics

Provide flowcharts highlighting the differences and similarities of the various procedures, especially parts which are identically or similarly shared as well as unique to each method. This may be achieved by modifying Figure 1 with highlights and augmenting it with word descriptions which clearly identify the specifics of the method, such as temperature, duration, sequence length, ratio of mixture, etc. 

A comparison table contrasting the different methods will also help

(3) High-resolution figures

Figure 3 - Must be printed with high-res/sharp contrast and with the different types of scars clearly highlighted

Figure 6 - All data should have an error bar. If that is not possible, please describe why.

The units on these figures are absolute. Please explain why it is not detailed in terms of ratio of mixtures, for e.g., number of transformants per …

(4) Discussions of broader impact

The overall goal of the paper should be made abundantly clear in the abstract, introduction and conclusion. At the moment it reads more like a report rather than an article. 

Other comments:

Provide reasons as to why the 3 chimeric primers are chosen?

Why are the units for the content of the mixture in mass (ng) as supposed ratio or concentration? For those units, which are normalized such as transformants/ng, it’s not clear whether this is chosen as supposed to transformants/(unit of mixture concentration)

Please succinctly provide more details on the various quantitation methods of the mixture in addition to mentioning the industrial kit used to perform the measurement

Do the authors mean any differences between recombination vs. amplification vs. transfection efficiency? Please expand

Author Response

Dear unknown reviewer:

Thanks to the sincere comments of the unknown reviewer, the paper has been revised thoroughly according to the comments and suggestions, please see the following point to point response.

Comments and Suggestions for Authors

The article describes a novel method for DNA amplification that is less erroneous in terms of nucleotide mutation, insertion, and deletion - collectively referred to as ‘scaring’ - but at the expense of recombination efficiency as a function of mixture concentration and sequence length. While the idea is interesting and the implementation legitimate, the paper severely lacks in

  • technical prose and clarity,

Thanks for the sincere suggestion. The paper has been revised in technical prose and clarity, as shown in highlighted red words

  • structures, including but not limited to schematics, flowcharts, and tables, for the many experimental procedures, many of details are similar or repeated,

Thanks for the sincere suggestion. The structures have been made more clear including schematics, flowcharts, and tables, for the many experimental procedures.

  • high-resolution figures with appropriate spacing and contrast, and

Thanks for the sincere suggestion. High-resolution figures have been made.

(4) discussions of the broader significance of the method within the commercial, academic, and healthcare communities. 

Thanks for the sincere suggestion. It is a good suggestion for commercial and healthcare. The discussion concentrates mainly on academic significance, because the authors serve as scientific researchers, and we are not familiar with commercial purpose.

The article shows promise that should be realized by giving it more thought to the four critical elements mentioned above, and thus, I do not as yet recommend this paper for publication. 

Below are a number of suggestions on how the article may be improved upon

(1) Technical prose

 Why is a particular DNA sequence, pET20b-AdD chosen for the experiment?

 Thanks for the sincere suggestion. Because T4 DNA ligase is widely used in DNA ligation (shown in revised paper as red words), and pET20b is widely used in protein expression, we used the DNA sequence as an example to construct plasmid.

What is specifically “convenient” or “inconvenient” about the different methods?

Thanks for the sincere suggestion.  “inconvenient” (shown in revised paper as red words) “by co-transforming inserts and vectors having homology arms, recovering recombined plasmids from clone host cells, and re-transforming them into expression host cells.” “convenient” as “recombination in vitro and transformation into expression host cells”

Define efficiency and accuracy when they are first mentioned.

 Thanks for the sincere suggestion. “efficiency and accuracy” have been defined in revised paper in red words.

Introduction should include current state-of-the-art DNA amplification method and with references to better define the relevance of the article

  Thanks for the sincere suggestion. “current state-of-the-art DNA amplification method and with references” has been included in revised paper in red words.

Conclusion should continue where introduction ends. 

 Thanks for the sincere suggestion. “Conclusion” has been revised to continue where introduction ends in red words.

Description of procedures, results and discussions should be clearly connected to each other even as they are segmented. These are not so in the article. Additionally, each section should flow from one to the next

 Thanks for the sincere suggestion. “Description of procedures, results and discussions” has been connected to each other in revised paper.

Pros and cons of each method are better summarized in the form of a table. 

Thanks for the sincere suggestion. Pros and cons of each method has been summarized in Table 1 in revised paper.

(2) Essential schematics

 Provide flowcharts highlighting the differences and similarities of the various procedures, especially parts which are identically or similarly shared as well as unique to each method. This may be achieved by modifying Figure 1 with highlights and augmenting it with word descriptions which clearly identify the specifics of the method, such as temperature, duration, sequence length, ratio of mixture, etc. 

 Thanks for the sincere suggestion. Figure 1 and legend have been revised to highlight the differences and similarities of the 3 methods in revised paper.

A comparison table contrasting the different methods will also help

 Thanks for the sincere suggestion. The comparison has been made in Table 1 in revised paper.

(3) High-resolution figures

 Thanks for the sincere suggestion. “High-resolution figures” have been revised

Figure 3 - Must be printed with high-res/sharp contrast and with the different types of scars clearly highlighted

 Thanks for the sincere suggestion. “Figure 3” have been revised with high-res/sharp contrast and with the different types of scars clearly highlighted.

Figure 6 - All data should have an error bar. If that is not possible, please describe why.

 Thanks for the sincere suggestion. Figure 6 - All data should have an error bar, the 3 other error bar were too small to appear.

The units on these figures are absolute. Please explain why it is not detailed in terms of ratio of mixtures, for e.g., number of transformants per …

 Thanks for the sincere suggestion. The units on these figures are revised as transformants/ng

(4) Discussions of broader impact

 The overall goal of the paper should be made abundantly clear in the abstract, introduction and conclusion. At the moment it reads more like a report rather than an article. 

 Thanks for the sincere suggestion. The overall goal of the paper is made more clear in the abstract, introduction and conclusion in the revised paper

Other comments:

 Provide reasons as to why the 3 chimeric primers are chosen?

 Thanks for the sincere comment. “Reasons as to why the 3 chimeric primers are chosen” has been added in red words in revised paper.

Why are the units for the content of the mixture in mass (ng) as supposed ratio or concentration? For those units, which are normalized such as transformants/ng, it’s not clear whether this is chosen as supposed to transformants/(unit of mixture concentration)

 Thanks for the sincere comment. All units are normalized as transformants/ng in revised paper.

Please succinctly provide more details on the various quantitation methods of the mixture in addition to mentioning the industrial kit used to perform the measurement

 Thanks for the sincere comment. Tansformation efficiency calculate as transformants/ng for all the 3 methods.

Do the authors mean any differences between recombination vs. amplification vs. transfection efficiency? Please expand

Thanks for the sincere comment. Tansformation efficiency is used in revised paper

Submission Date

18 Feb 2020

Reviewer 2 Report

The authors describe the use of Exnase II in restriction enzyme-free cloning of inserts into plasmids. They show that their method leads to a higher frequency of scarless plasmids then the Exnase cloning or Two-step PCR / ligase method.

Due to the poor English, it took me a while before I was able to understand what the message of the paper was. In the introduction more information about Exnase and ExnaseII is necessary. In figure 1 in the left, both Exnase and Exnase II are mentioned. Is this true or is it ExnaseII in both cases.

The text contains a lot of repetitions. The main results in the body text are repeated in the legend of the figures. Due to the layout, it is difficult to see what part of the text belongs to the legends and what belongs to the body of the text.

Is the scarless-plasmid the result of Exnase II or is it because only one recombination site was used in the intra method? What happens if Exnase and Exnase II were exchanged in the two inter and intra methods? So Exnase in intra-HR and ExnaseII in inter-HR. The results can tell if this is an enzyme related property or that it is the specific design of the sequences in the overlapping regions.

I do not recommend accepting the manuscript in its current form. I think there must be a better explanation of why ExnaseII is better than Exnase before the majority of the molecular biologists are willing to accept ExnaseII intra-HR cloning.

Below you can find some examples of errors.

poor quality of the figures (resolution is too low)

line 52, 61: PCP; PCR

line 52: cloing; cloning

line 59: arrow star: I could not find an arrow star

figure 1: red box around sequence contains a black vertical line. What is this?

line 111,112,163: dimmer; dimer

fig 4: Absolute number of transformants or transformants / ng ?

fig 4b: ExnaseII in the intra-HR part?

line 152,167: regressed; correlated

line 221, 222: no colored lines are visible in the table.

Author Response

Dear unknown reviewer:

Thanks to the sincere comments of the unknown reviewer, the paper has been revised thoroughly according to the comments and suggestions, please see the following point to point response.

Comments and Suggestions for Authors

The authors describe the use of Exnase II in restriction enzyme-free cloning of inserts into plasmids. They show that their method leads to a higher frequency of scarless plasmids then the Exnase cloning or Two-step PCR / ligase method.

Due to the poor English, it took me a while before I was able to understand what the message of the paper was. In the introduction more information about Exnase and ExnaseII is necessary. In figure 1 in the left, both Exnase and Exnase II are mentioned. Is this true or is it ExnaseII in both cases.

Thanks to the sincere comments. English has been improved in revised paper.  “more information about ExnaseII” has been added as “a commercial recombinase ExnaseII” in revised paper. “In figure 1 in the left” it is ExnaseII in both cases in revised paper.

The text contains a lot of repetitions. The main results in the body text are repeated in the legend of the figures. Due to the layout, it is difficult to see what part of the text belongs to the legends and what belongs to the body of the text.

Thanks to the sincere comments. “a lot of repetitions” have been deleted in revised paper. the legend of the figures” has been revised and write in smaller words to differ from that of text in revised paper.

Is the scarless-plasmid the result of Exnase II or is it because only one recombination site was used in the intra method? What happens if Exnase and Exnase II were exchanged in the two inter and intra methods? So Exnase in intra-HR and ExnaseII in inter-HR. The results can tell if this is an enzyme related property or that it is the specific design of the sequences in the overlapping regions.

Thanks to the sincere comments. the scarless-plasmid” is the result of only one recombination site used in the intra method. Exnase is Exnase II.  

I do not recommend accepting the manuscript in its current form. I think there must be a better explanation of why ExnaseII is better than Exnase before the majority of the molecular biologists are willing to accept ExnaseII intra-HR cloning.

Below you can find some examples of errors.

poor quality of the figures (resolution is too low)

Thanks to the sincere comments.the figures have been revised in high resolution.

line 52, 61: PCP; PCR

Thanks to the sincere comments.“PCP” has been changed to “PCR”

line 52: cloing; cloning

Thanks to the sincere comments.cloing” has been changed to “cloning”

line 59: arrow star: I could not find an arrow star

Thanks to the sincere comments. The word deleted in revised paper.

figure 1: red box around sequence contains a black vertical line. What is this?

Thanks to the sincere comments.black vertical line shows “the scarless junction” in revised paper.  

line 111,112,163: dimmer; dimer

Thanks to the sincere comments.dimmer” has been changed to “dimer”in revised paper.

fig 4: Absolute number of transformants or transformants / ng ?

Thanks to the sincere comments. Absolute number of transformants”has been changed to “transformants / ng” in revised paper.

fig 4b: ExnaseII in the intra-HR part?

Thanks to the sincere comments. Fig4b has been intetrated in Fig1 in revised paper. ExnaseII used in the intra-HR and inter-HR.

line 152,167: regressed; correlated

Thanks to the sincere comments. regressed” has been changed to “correlated” in revised paper.

line 221, 222: no colored lines are visible in the table.

Thanks to the sincere comments. colored lines” has been changed to “underlined line”in revised paper.

Submission Date

18 Feb 2020 16:34:18

Round 2

Reviewer 1 Report

Dear authors,

After further review, I am now recommending this paper for publication. I nevertheless still have the following suggestions:

  1. The paper, while has been greatly improved in its writing and numerous descriptions per the reviewers’ comments, still suffers from cumbersome English. And while I understand the authors’ unfair difficulty in this regard, I strongly suggest - if at all possible - hiring or involving an English language editor/writer. This is to do the authors’ communication of their good work as well as the journal quality justice. 
  2. Table 1 can now be more appropriately situated near the conclusion as they include the results post materials and methods section.
  3. Now that it is clear that the main difference between the two-step PCR and intra-HR is the use of Exnase vs. DNA ligase post plasmid amplification, would the author please expand on why the former seems to lead to more nc-plasmid and in turn higher transformant efficiency than the latter? 
